# Evaluation of Pulp Repair after Biodentine^TM^ Full Pulpotomy in a Rat Molar Model of Pulpitis

**DOI:** 10.3390/biomedicines9070784

**Published:** 2021-07-06

**Authors:** Sandra Minic, Marion Florimond, Jérémy Sadoine, Anne Valot-Salengro, Catherine Chaussain, Emmanuelle Renard, Tchilalo Boukpessi

**Affiliations:** 1Laboratory of Orofacial Pathologies, Imaging and Biotherapies, School of Dentistry, Laboratoire d’excellence INFLAMEX, Université de Paris, URP 2496, 1 Rue Maurice Arnoux, 92120 Montrouge, France; sandra.minic16@gmail.com (S.M.); marion.florimond@live.com (M.F.); sadoine.j@gmail.com (J.S.); catherine.chaussain@u-paris.fr (C.C.); 2AP-HP Department of Dental Medicine, Charles Foix and Bretonneau Hospitals, and Reference Center for Rare Diseases of Calcium and Phosphorus Metabolism, 7 Avenue de la République, 94200 Ivry-sur-Seine, France; 3Septodont SAS, 105 Avenue Beaurepaire, 94100 Saint-Maur-des-Fossés, France; avalotsalengro@septodont.com; 4Inserm, UMR 1229, RMeS, Regenerative Medicine and Skeleton, Université de Nantes, ONIRIS, 1 Place Alexis Ricordeau, 44000 Nantes, France; 5CHU de Nantes, Service d’Odontologie Restauratrice et Chirurgicale, 1 Place Alexis Ricordeau, 44000 Nantes, France

**Keywords:** pulpal inflammation, dentin-pulp complex regeneration, vital pulp therapy, reparative dentin, calcium silicate-based cement, animal model

## Abstract

Dental pulp is a dynamic tissue able to heal after injury under moderate inflammatory conditions. Our study aimed to evaluate pulp repair under inflammatory conditions in rats. For this purpose, we developed a rat model of controlled pulpitis followed by pulpotomy with a tricalcium silicate-based cement. Fifty-four cavities were prepared on the occlusal face of the maxillary upper first molar of 27 eight-week-old male rats. *E. coli* lipopolysaccharides at 10 mg/mL or phosphate-buffered saline PBS was injected after pulp injury. Non-inflamed molars were used as controls. Levels of inflammation-related molecules were measured 6 and 24 h after induction by enzyme-linked immunosorbent assay of coronal pulp samples. Pulp capping and coronal obturation after pulpotomy were performed with tricalcium silicate-based cement. Four and fifteen days after pulpotomy, histological and immunohistochemical analysis was performed to assess pulp inflammation and repair processes. Our results showed significantly higher levels of innate inflammatory proteins (IL-1β, IL-6, TNF-α and CXCL-1) compared with those in controls. Moderate residual inflammation near the capping material was demonstrated by histology and immunohistochemistry, with the presence of few CD68-positive cells. We showed that, in this model of controlled pulpitis, pulpotomy with Biodentine^TM^ allowed the synthesis at the injury site of a mineralized bridge formed from mineralized tissue secreted by cells displaying odontoblastic characteristics. Analysis of these data suggests overall that, with the limitations inherent to findings in animal models, pulpotomy with a silicate-based cement is a good treatment for controlling inflammation and enhancing repair in cases of controlled pulpitis.

## 1. Introduction

Dental pulp (DP) is a highly dynamic connective tissue encapsulated in mineralized dental tissues, which is responsible for tooth vitality, sensitivity, immune response and also repair and regeneration. Its network of resident immunocompetent cells plays a significant role in the defense against pathogens during tissue injury [1,2,3]. Inflammation of DP tissue, called pulpitis, is one of the most common dental disorders characterized by a tightly regulated sequence of vascular and cellular events mediated by molecular factors such as cytokines and chemokines [4]. Indeed, in response to pathogen-associated molecular patterns (PAMPs), such as lipopolysaccharides (LPS), DP cells upregulate innate immunity effectors, leading to the recruitment and activation of immune cells [5]. These inflammatory events mediate the interplay between infection, host defense, tissue injury and repair [6]. Depending on the extent of dentin–pulp complex injury, the biological response will be reactionary or reparative. During reparative dentinogenesis, the cellular responses are much more challenging and require the recruitment of stem/progenitor cells to the injured site and their subsequent proliferation and differentiation into a new generation of odontoblast-like cells [7].

In clinical practice, pulpitis is diagnosed as reversible or irreversible. Although this classification is well recognized, its applicability has recently been questioned [8]. Further, the limitations of current diagnostic methods are widely acknowledged [9] and mean that there is great interest in identifying and assessing potential biomarkers of pulp disease as diagnostic tools [10,11,12], with a view to making vital pulp therapy (VPT) outcomes more reliable.

VPT procedures consist of direct pulp capping, or partial or full pulpotomy with bioactive capping materials. Typical bioactive materials used in pulpal and other endodontic procedures to enhance healing include the family of tricalcium silicate-based (TCS-based) cements. Several commercial TCS-based cements are available with subtle differences in their manufacturing process and composition. ProRoot White MTA (Dentsply, Tulsa Dental, Tulsa, OK, USA) and Biodentine^TM^ (Septodont, Saint-Maur-des-Fosseés, France) are two representative TCS-based cements with great clinical success in dentistry [13]. The main reason these TCS-based cements are widely recommended in clinical practice is that they are able to form calcium hydroxide, a by-product of the hydration process. The subsequent dissolution of calcium hydroxide releasing hydroxide (OH-) and calcium ions (Ca^2+^) creates an environment favorable to healing and repair of soft and hard tissues [14,15]. On the one hand, hydroxide ions create an alkaline environment which has an antibacterial and anti-inflammatory effect, and on the other, Ca^2+^ ions play a major role as intracellular second messengers that regulate a wide range of cellular processes, including gene transcription; protein expression; cell proliferation, differentiation and apoptosis; and excitatory cell activation [16,17]. Recently, Biodentine^TM^ has been shown to induce the release of large numbers of Ca^2+^ ions, this being reflected in altered intracellular Ca^2+^ dynamics, and, in turn, differential gene expression, cellular differentiation and mineralization potential of human dental pulp stem cells (hDPSCs) stimulated with TCS-based cements [18,19].

Recent reviews have reported good outcomes with the use of the aforementioned products in terms of DP healing and a wide range of success rates for pulp capping procedures after carious exposure [20,21,22]. This wide range seems to be attributable to differences in the level of pulp inflammation, control of infection and bioactivity of capping materials. Therefore, to optimize outcomes, indication for VPT should be based on an in-depth understanding of how inflammatory pulp disease can modify pulp healing responses, and for this, there is a need to model the interplay of infection and inflammation with injury and reparative events. Accordingly, in the present study, we aimed to develop an LPS-inflamed controlled pulpitis model on rat molars and to evaluate pulp repair after pulpotomy with a TCS-based cement.

## 2. Materials and Methods

### 2.1. Animals

A total of 27 eight-week-old male Sprague Dawley rats (Janvier Labs, Saint-Berthevin, France) were used for the study. The animal experiment was designed and is reported in accordance with the ARRIVE guidelines and EU Directive 2010/63/EU for animal experiments and was approved by the Animal Care Committee of Université de Paris (protocol No. C92-049-01/ APAFiS no.19-042).

### 2.2. Induction of Pulpitis

Induction of pulpitis was performed under general anesthesia with an intraperitoneal injection of 10% ketamine (Imalgene 500, Merial, France) and 2% xylazine (Rompun, Bayer, Lyon, France). Aided by an operative microscope (Carl Zeiss, Oberkochen, Germany), 54 cavities were prepared on the occlusal surfaces of the maxillary right and left first molars, using a 0.6 mm-diameter round carbide bur (Dentsply Maillefer, Ballaigues, Switzerland). Pulpitis was induced by LPS from *Escherichia coli* as described by Renard et al. [3]. Thus, after the pulp was mechanically exposed, 5 µL of LPS from *Escherichia coli* O111:B4 (Sigma Chemical Co., St. Louis, MO, USA) at a final concentration of 10 mg/mL or phosphate-buffered saline (PBS) was applied. The cavities were sealed with Cavit (ESPE, Seefeld, Germany). Then, the rats were divided into two groups depending on induction time (6 or 24 h) (Figure 1).

### 2.3. Pulpotomy Procedure

Six or 24 h after pulpitis induction, the pulp was exposed again. The pulp tissue in the crown was harvested using sterile paper tips, transferred to a micro-centrifuge tube (Biopur 1 mL, Vaudaux-Eppendorf AG, Schönenbuch, Switzerland) and immediately stored at −80 °C until use. After achievement of hemostasis with sterile cotton pellets, Biodentine^TM^ cement was used to fill the DP coronal space, as per the manufacturer’s guidelines (Figure 1). At day 4 (D4) or day 15 (D15) after pulpotomy, rats were sacrificed. Intracardiac perfusion was performed using 4% paraformaldehyde buffered with 0.1 M of sodium cacodylate at pH 7.2–7.4. Hemi-maxillae were removed, dissected and fixed with 4% paraformaldehyde for 24 h at 4 °C. Figure 1 outlines the experimental procedure and distribution of animals.

### 2.4. Enzyme-Linked Immunosorbent Assay

Multiplex assays (Meso Scale Discovery [MSD], Rockville, MD, USA) were used to determine cytokine concentrations produced by LPS/PBS injection into harvested pulp samples. Paper tips were placed in a solution containing a lysis buffer (pure Tris, CaCl2, 0.9% NaCl and 0.2% Triton) and a protease cocktail (Protease Inhibitor Cocktail Set V EDTA-Free, Calbiochem^®^, Merck, Darmstadt, Germany). The MSD V-Plex Proinflammatory Panel 2 Rat Kit was used to measure the total protein concentrations of interleukin-1 beta, -4, -5, -6, -10 and -13 (IL-1β, IL-4, IL-5, IL-6 IL-10 and IL-13), chemokine ligand 1 (CXCL-1), tumor necrosis factor-alpha (TNF-α) and interferon-gamma (IFN-γ) in samples, according to the manufacturer’s instructions.

### 2.5. Sample Preparation for Histology

Decalcification was performed in a solution of 4.13% EDTA (pH 7.4) maintained at 45°C with a Milestone KOS Microwave Tissue Processor (Sorisole, Bergamo, Italy) for approximately 4 weeks. Tissue blocks were then embedded in Paraplast and 5 μm thick sections were cut with a microtome (Microm HM 325, Microm Microtech, Brignais, France) along the mesiodistal axis.

### 2.6. Histological Analysis

Sections were stained with either hematoxylin and eosin or Masson’s trichrome to assess inflammation and characterize the mineralized bridge. For histological analysis, the samples were examined with a microscope (Leica, Nanterre, France). Four observers, blinded to group allocation and type of section, examined the slices and evaluated the inflammatory state of the pulp and the mineralized barrier formed with a scoring system as reported in a previous study [23] (Table 1, Table 2 and Table 3). Scores analyzed were agreed on by at least three of the four observers.

### 2.7. Immunohistochemistry

For immunohistochemical analysis, sections were incubated with primary antibodies, a mouse monoclonal CD68 (clone ED1) (MAB1435, Merck KGaA, Darmstadt, Germany) diluted at 1/100 and a mouse monoclonal dentin sialoprotein (DSP; LF-153, kindly donated by Larry Fisher) at 1/200. Tissue sections were then incubated with secondary antibodies at 1/200 Rhodamine Red-X goat anti-mouse IgG (H+L) (Thermo Fisher Scientific, Waltham, MA, USA) for CD68 and Alexa Fluor 488 rabbit anti-mouse IgG (H+L) (Thermo Fisher Scientific) for DSP. The negative control was obtained by replacing the primary antibody with the mouse pre-immune serum. For nucleus staining, we used DAPI from Thermo Fisher Scientific.

### 2.8. Micro-CT Follow-Up

The micro-computed tomography (micro-CT) analysis was performed with a Quantum FX μCT scanner (PerkinElmer, Waltham, MA, USA), allowing image resolution ranging from 10 to 295 μm using low doses of radiation. A 30 mm field of view was used, and the reconstruction was performed with NRecon software (Skyscan, Aartselaar, Belgium). The 3D images acquired were analyzed with Osirix DICOM viewer software (version 3.7.1) in multiplanar reconstruction mode.

A longitudinal follow-up was carried out, with measurements taken before and after the induction, after pulpotomy and at D4 or D15 (immediately before sacrifice). Representative images are shown in Figure 1 and Figure 4. Mineralization was assessed in 3D sections of several teeth. Roots were analyzed one by one with CTAn software (Skyscan).

### 2.9. Statistical Analysis

All quantitative data are presented as the mean ± SD. Statistical analysis for enzyme-linked immunosorbent assay was performed by one-way analysis of variance with Tukey’s post hoc test. Histological analysis was performed with Student’s t test. Mineralization of the roots was statistically analyzed with the Kruskal–Wallis test using Prism software. Differences were considered significant at *p* < 0.05.

## 3. Results

### 3.1. Cytokine Profile in the Inflamed Coronal Pulp Chamber

The inflammation induced by PBS or LPS was evaluated by comparing the pulp concentrations of a range of biological markers at the time of pulpotomy to those in healthy pulp (control). The protein expression levels of IL-1β, IL-6, TNF-α, CXCL-1, IL-10, IL-4, IL-5, IL-13 and INF-γ are presented in Figure 2. Notably, the concentrations of IL-1β, IL-6, CXCL-1 and TNF-α were significantly higher in the inflamed pulp (PBS or LPS) than in the control pulp. Nevertheless, no significant differences were found between the PBS and LPS groups. The quantitative measurements of IL-4, IL-5, IL-13 IL-10 and IFN-γ did not reveal any significant differences between the three groups or between the two induction times used.

### 3.2. Residual Pulp Inflammation after Pulpotomy at D4 and D15

Representative examples of hematoxylin-eosin staining and immunofluorescence on LPS/PBS-induced pulpitis (6 and 24 h) and healthy pulp at D4 and D15 are shown in Figure 3. Table 1 outlines the inflammatory reaction scoring system used, and Table 2 outlines the scores obtained at D15 after pulpotomy. On D15, the percentage of DP with no observable inflammation was significantly higher for the 6-h than the 24-h induction time. On the other hand, the score for the second criterion “Mild: inflammatory cells only next to dentin bridge or area of pulp exposure” was significantly higher for the 24-h than the 6-h induction. Differences in histological appearance between tissue exposed to LPS and PBS did not reach significance.

Inflammation was also characterized at the cellular level. Specifically, the presence of monocytes was assessed by immunofluorescence using CD68 labelling. Four days after pulpotomy, we observed more CD68+ cells after 24-h (Figure 3h,i) than 6-h induction, regardless of the stimulant (LPS or PBS) (Figure 3f,g). Labelling was found mainly in the coronal third of the root, very close to the cellular degeneration area. Fifteen days after pulpotomy, the inflammation had resolved in the pulp with 6-h induction (LPS/PBS) (Figure 3p,q) and healthy pulp (Figure 3t). A few macrophage cells remained in the pulp that had undergone 24-h induction of inflammation by PBS or LPS exposure (Figure 3r,s).

**Table 1 biomedicines-09-00784-t001:** Scoring table used for inflammation analysis, on hematoxylin and eosin-stained histological sections 15 days after pulpotomy; see representative sections in Figure 3k–o (LPS/PBS, 6/24 h of induction).

Score
Inflammatory reaction None Mild: inflammatory cells only next to dentin bridge or area of pulp exposure Moderate: inflammatory cells are observed in part of radicular pulp Severe: all coronal pulp is infiltrated or necrotic General state of the pulp No inflammatory reaction Inflammatory reaction Abscess Necrosis

**Table 2 biomedicines-09-00784-t002:** Evaluation of inflammation on hematoxylin and eosin-stained histological sections on D15 using the scoring table (Table 1). A *t*-test was performed to detect any significant differences between the groups using GraphPad Prism version 6 (GraphPad Software, CA, USA) * *p* < 0.05. Results for 6-h PBS and LPS treatment were significantly different from those for 24-h PBS and LPS treatment for criteria 1 and 2 of inflammatory reaction and criteria 1, 2 and 3 for the general state of pulp.

Inflammatory Condition Groups	Product Injected	Inflammatory Reaction				General State of the Pulp			
		1	2	3	4	1	2	3	4
Control		25%	50%	25%	0%	50%	25%	25%	0%
6 h	PBS	42.85% *	28.57% *	0%	28.57%	28.57% *	57.14% *	14.28% *	0%
pulpitis induction	LPS	33.33% *	33.33% *	33.33%	0%	33.33% *	66.66% *	16.66% *	0%
24 h	PBS	27.27%	45.45%	27%	0%	18.18%	72.72%	9.09%	0%
pulpitis induction	LPS	22.22%	55.55%	22.22%	0%	22.22%	77.77%	0%	0%

### 3.3. Characterization of the Mineralized Bridge at D15

At D15 after pulpotomy, micro-CT analysis of the treated teeth revealed the formation of a reparative mineralized bridge in the residual pulp in all the groups (Figure 4a). This mineralized barrier was in contact with the lateral dentin walls and closed the opening of the root canal. In addition, we noticed the presence of mineralized tissue in the pulp canal space in the 24-h induction group (LPS or PBS) (Figure 4a,c,d,h,i,m,n). Moreover, quantifying the mineralization of each root of all teeth (Figure 4b), we noticed a tendency of less mineralization in inflamed pulp than in healthy pulp without statistical significance. No difference in mineralization between the 24 h and 6 h was observed.

Then, we histologically examined sections stained with Masson’s trichrome, rating pulp-dentin repair with the scores set out in Table 3. Representative examples of the characteristic appearance observed for each condition are shown in Figure 5. The percentage of roots in each state were summed and the scores are reported in Table 4. At D15 after pulpotomy, a newly mineralized barrier was observed in most specimens in all the groups. More specifically, considering the morphological score, the percentage of roots for which the mineralized bridge showed “only irregular hard tissue deposition” was higher in the 6-h than the 24-h induction groups, while the percentage showing “dentin or dentin-associated with irregular hard tissue” (criterion 1) was higher in the 24-h than the 6-h induction groups. Further, the location of the new mineralized tissue varied between the conditions. The percentage of roots for which the calcified barrier was “close to the exposed area without invading the radicular space” was higher in the 6-h induction groups (LPS and PBS), while the percentage showing mineralization “invading more than 50% of the radicular area” was higher in the 24-h induction groups. We did not find differences according to stimulant (LPS or PBS). A thin necrotic layer between the new mineralized barrier and the pulp capping agent was observed in all groups (Figure 5).

**Table 3 biomedicines-09-00784-t003:** Scoring table used for mineralization analysis on Masson’s trichrome histological sections on D15; representative sections presented in Figure 5a–j (LPS/PBS and 6/24 h conditions).

Score	
Hard Tissue	
1	Present
4	Absent
Continuity	
1	Complete
2	Little contact of the capping material with dental pulp
3	Only lateral deposition of hard tissue on radicular pulp
4	No hard tissue bridge or lateral deposition of hard tissue
Morphology	
1	Dentin or dentin associated with irregular hard tissue
2	Only irregular hard tissue deposition
3	Only a thin layer of hard tissue deposition
4	No hard tissue deposition
Localization	
1	Close to the exposed area without invading the radicular space
2	Bridge invading pulp space up to 50% of radicular space
3	Bridge invading more than 50% of radicular space
4	No bridge or only hard tissue deposition on walls of the exposed cavity

**Table 4 biomedicines-09-00784-t004:** Characterization of the mineralization reaction on Masson’s trichrome stained histological sections using the scoring table (Table 3). A *t*-test was performed to detect the significant differences between the groups using GraphPad Prism version 6 (GraphPad Software, CA, USA) * *p* < 0.05. Results for 6-h PBS and LPS treatment were significantly different from those for 24-h PBS and LPS treatment for criteria 1 and 2 of the morphology description and criteria 1 and 3 for localization specificities.

Inflammatory Condition Groups	Product Injected	Hard Tissue		Continuity				Morphology				Localization			
		1	4	1	2	3	4	1	2	3	4	1	2	3	4
Control		75%	25%	50%	0%	25%	25%	0%	25%	50%	25%	25%	50%	0%	25%
6 h pulpitis	PBS	71.40%	28.60%	14.28%	14.28%	42.85%	28.57%	0% *	42.85% *	28.57%	28.57%	14.28% *	28.57%	28.57% *	28.57%
induction	LPS	83.33%	16.60%	33.33%	33.33%	16.66%	16.66%	16.66% *	33.33% *	16.66%	16.66%	16.66% *	50.0%	0% *	16.66%
24 h pulpitis	PBS	73%	27%	18.18%	9.09%	45.45%	27.27%	27.27%	18.18%	27%	27%	0%	9.09%	63.63%	27.27%
induction	LPS	89%	11.11%	55.55%	0%	33.33%	11.11%	33.33%	22.22%	33.33%	11.11%	0%	33.33%	55.55%	11.11%

### 3.4. Characterization of Cells Involved in DP Repair

Four and fifteen days after pulpotomy, we performed immunofluorescence labelling with DSP to determine whether cells involved in DP repair had odontoblastic characteristics. At D4, for 6- and 24-h of LPS or PBS exposure, there were few DSP+ cells labelled in green, and those present were mostly concentrated close to the tissue/material interface (Figure 6a,d), while some other DSP+ cells were detected near the odontoblastic layer along with root dentin.

Fifteen days after pulpotomy, all the induction conditions displayed DSP labelling. Labelling extended over a wider area for 24-h (Figure 6h,i) than for 6-h (Figure 6f,g) induction with PBS/LPS, and cells were located near the mineralized barrier at the canal opening but also along the root wall.

## 4. Discussion

This in vivo study shows for the first time that a calcium silicate-based cement induces dentin repair when applied as pulp capping material after pulpotomy in a controlled pulpitis model in rat molars. Our study is novel in that most DP studies on rat molars have been performed after pulpotomy in healthy conditions [24,25]. In addition, our study is original in that in most cases of experimental induced pulpitis in rats, the studies were realized on incisors. However, rodent incisors are continuously erupting teeth with high healing potential after pulp injury, thus removing them from human clinical reality [26,27].

### 4.1. Development of a New Model of Pulp Inflammation in Rats

Pulpal inflammation under caries is elicited by bacterial antigens that diffuse into the pulp through dentinal tubules. One of these components, LPS, a major outer membrane component in Gram-negative bacteria detected in deep-seated dental caries [28], is associated with the pathogenesis of pulpitis. It is known that this PAMP stimulates toll-like-receptor 4 on DP cells and activates the NF-kB pathway, producing inflammatory cytokines such as IL-1β and IL-6 [29,30]. Nonetheless, the relationships among LPS, inflammatory cytokines and hard tissue formation in DP remain unclear. In this study, we evaluated the ability of DP to heal after stimulation with LPS. LPS-induced dental pulp inflammation represents a stable experimental inflammation model for the dynamic observation of the progress of acute and chronic pulpitis [27]. We chose LPS from Escherichia coli for its ability to stimulate the TLR4 receptor, an ability similar to Gram-negative bacteria found in deep caries [3,26,31,32]. This model is an innate inflammatory model; indeed, we show that the expression of several innate inflammatory proteins, namely, IL-1β, IL-6, TNF-α and CXCL-1, is significantly higher at 6 or 24 h after inflammation induction than in healthy pulp, but this is not the case for the expression of IL-4, IL-5, IL-13 or INF-γ, which are proteins more involved in the adaptative immune response [33].

Interestingly, 6 or 24 h after inflammation induction, we did not detect any change in the expression of IL-10, a cytokine involved in the regulation of inflammation. In contrast, IL-10 was shown to be upregulated at 3 h in a rat incisor LPS-induced inflammation model, the difference disappearing at 9 h [3]. This absence of IL-10 overexpression at later time points suggests that the inflammatory process may already be in the resolution phase, favorable to repair.

Our study also found no difference in expression as a function of the stimulant, LPS or PBS. Few studies have explored the effects of LPS in rat molar DP; exceptions include one that investigated the response to LPS injection into non-inflamed pulp [34] and another that compared LPS injection to injury, finding that LPS appeared to be acting on Notch signaling synergistically with mechanical damage [35]. Other studies have shown that the irritation of the DP by drilling a hole in dentin and without exposure to bacteria on rat incisors was sufficient to induce an increase in the number of white blood cells (granulocytes and lymphocytes) [33,36,37].

In our study, we detected large numbers of CD68+ monocytes throughout the radicular pulp and particularly in the coronal part at D4, with no differences according to stimulant (LPS/PBS) or with respect to control or by induction duration, but detected far fewer of these cells at D15 in treated animals. CD68 is considered a marker for macrophage lineage cells, and macrophages are immune cells involved in the elimination of pathogens and tissue homeostasis through the clearance of senescent cells and repair after inflammation. Further, it is known that CD68+ monocytes are resident in the DP [38], and their density increases with the progression of caries [39]. Overall, we consider that the reduction in CD68+ cells observed at the later follow-up is suggestive of resolution of the inflammation.

### 4.2. Biodentine^TM^ Induces Pulp-Dentin Complex Regeneration in Inflammatory Conditions

In the present study, to evaluate pulp repair, we used Biodentine^TM^, a resin-free material, which is mainly composed of pure TCS and contains calcium chloride, as a setting accelerator. The TCS family is now considered the material of choice for VPT since it induces dentin-pulp repair without pulpal inflammation, as has been demonstrated in vivo in a rat model [40] and in human teeth [41]. Recently, Giraud et al. showed in vitro that Biodentine^TM^ decreased inflammation and induced the secretion of growth factors involved in the healing process [42]. Other recent clinical data have demonstrated that Biodentine™ and MTA^®^ used as pulp capping to treat irreversible pulpitis leads to pulp function restoration and/or dentin bridge formation in immature and mature teeth [43,44]. As noted above, TCS-based cements are widely recommended in clinical practice because of their ability to form calcium hydroxide.

Indeed, most TCS-based cements lead to the formation of calcium hydroxide and leaching of hydroxyl and calcium ions as demonstrated for MTA^®^ and Biodentine^TM^, amongst others [45,46]. The hydroxyl ions released upon hydration increase the pH in the underlying tissue leading to a thin necrotic layer between the remaining vital tissue and the pulp capping agent, as we have shown in this study [47,48]. The presence of a necrotic zone protects the underlying vital pulp cells from the material’s alkaline pH. Furthermore, it allows the underlying pulp cells to participate in repair and regeneration processes [49]. The alkaline pH also has an anti-microbial effect [50]. Subsequent calcification of this superficial necrotic layer, followed by tertiary dentin formation from stimulated and differentiated dental pulp stem cells, gives rise to a protective dentin bridge [49]. Calcium ions are involved in dentin bridge formation through the stimulation of DPSC differentiation [51] and the formation of mineralized matrix nodules [51,52].

In our study, the new calcified tissue formed in inflamed pulp 6 h after induction was found close to pulp exposure or invading less than 30% of the pulp space, similar to the pattern observed in the control group and comparable to the mineralized bridge observed after direct pulp capping. That is, the 6-h induction time appears compatible with the formation of an efficient mineralized bridge after pulpotomy with Biodentine^TM^.

The impaired mineralization observed after the 24-h induction process suggests that the inflammation lasted long enough to disrupt the repair process but not enough to induce necrosis. Indeed, the presence of cellular inclusions evokes osteodentin formation (Cox et al., 1996). We may also suppose that the mineralization process is in progress. Connective tissue has been shown to be present in the mineralized barrier shortly after pulpotomy due to initial disorganization of the reparative dentin; over time, however, the mineralized wall becomes denser and more organized [53]. Considering this, the resolution of inflammation and the repair process could take longer in pulp with bacterial inflammation, which mimics caries, than in healthy DP. Fifteen days after pulpotomy would, therefore, be too short to observe the formation of a complete mineralized barrier. Difficulties found with the resistance of rat molar after 15 days and the higher risk of fracture in the longitudinal follow-up indicate the limits of this model and may warrant moving to a larger animal model.

Furthermore, the presence of mineralization extending along the canal after the 24-h induction process is consistent with the observation by other authors of radicular dentin in inflamed human teeth [54]. They reported the presence of atubular tertiary dentin on the root canal walls in teeth healed after VPT. The presence of cells immunolabelled with DSP suggested that cells secreting this structure had certain odontoblastic characteristics. DSP is considered a specific marker of functional odontoblasts due to its role in dentin initiation, formation and regulation [55]. On the other hand, a lack of unique molecular or morphological markers for physiological odontoblasts and the changing nature of these cells through their life cycle [56] severely hinder the identification of odontoblast-like cells. Though expression of dentin-sialophosphoprotein, DSP, nestin and several other markers is a recognized part of the molecular profile of odontoblasts, they certainly cannot be considered definitive markers of this cell phenotype [57].

Although there are several cements belonging to the TCS family, we chose to use Biodentine^TM^ for this first part of our research aiming to evaluate pulp repair under inflammatory conditions. We plan to compare the repair process according to various biomaterials from the TCS family in later work.

### 4.3. A new model of Pulp Inflammation Suitable for Pulp Repair Studies

Overall, with the limitations inherent to findings in animal models, our data suggest that an environment of pulpotomy together with induced controlled inflammation is suitable for the resolution of inflammation of the remaining radicular pulp. Our study failed to create a level of pulpal inflammation such that repair was no longer possible. Achieving such a level of inflammation is the next step in the long road to a better understanding of pulp biology, and it may be feasible by building on the controlled inflammation model described herein. Specifically, it would be interesting to increase the level of inflammation to reach the stage of irreversible pulpitis and then study repair after pulpotomy with a suitable capping material or natural molecules. Such approaches will make it possible to generate scientific evidence supporting the clinical application of techniques for the preservation of pulp vitality in cases of irreversible pulpitis. Thus, this animal model opens very interesting avenues for future research.

## Figures and Tables

**Figure 1 biomedicines-09-00784-f001:**
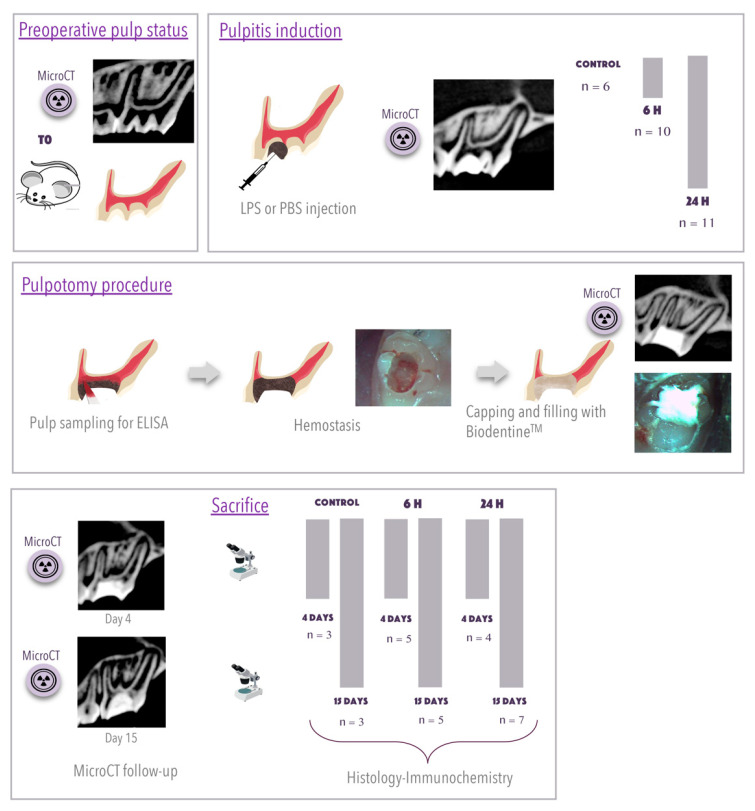
Experimental diagram and distribution of animals during the study. The first box represents the micro-computed tomography (micro-CT) exam of the preoperative pulp status. The second box details the induction of pulpitis after drilling a hole in the two maxillary first molars and the injection of 5 µL of phosphate-buffered saline (PBS) or lipopolysaccharide (LPS) into each tooth. The diagram indicates the division of animals by duration (0, 6 or 24 h) of pulpitis induction. The third box illustrates the pulpotomy procedure and pulp sampling with sterile paper tips for enzyme-linked immunosorbent assay analysis. Hemostasis (represented with a schematic drawing and a photograph) was achieved by injecting sterile saline and applying slight pressure with a sterile cotton pellet. Then, Biodentine^TM^ cement was used to fill the coronal pulp space. A micro-CT exam confirmed the postoperative result. The final box details the distribution of animals by treatment duration before sacrifice, 4 or 15 days, for histological and immunochemical analysis. A final micro-CT exam before sacrifice allowed verification of the impermeability of the obturation. All 27 animals were included in the analysis; no animals were excluded. The sample size was decided using the resource equation method (PMID 24250214), and an error degree of freedom of 22 was considered as indicating an acceptable sample size. Rats were randomly assigned to one of the groups. A numbering strategy was used to minimize potential confusion of animals, while permitting a level of blinding of operators during the experiment.

**Figure 2 biomedicines-09-00784-f002:**
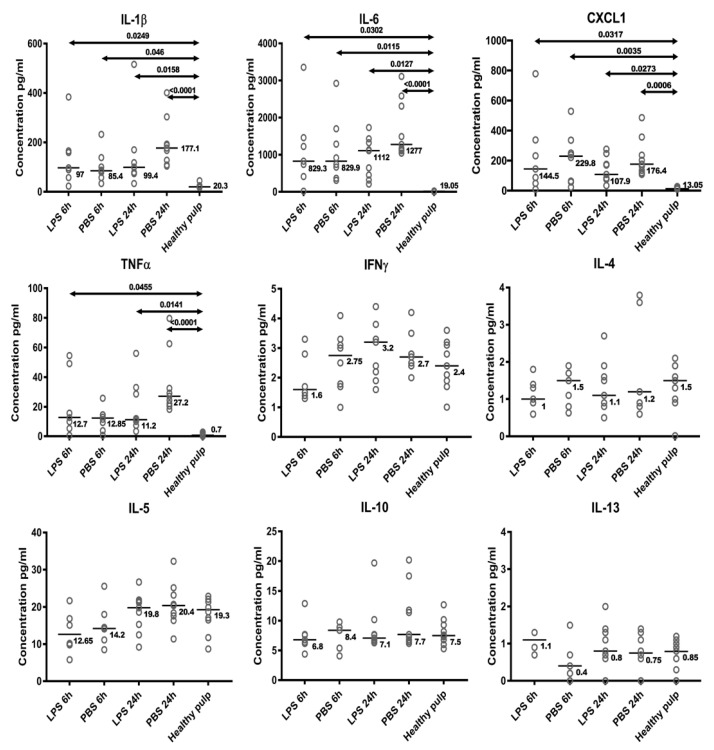
Analysis of expression levels of inflammatory proteins (interleukin-1 beta, -4, -5, -6, -10 and -13), chemokine ligand 1 (CXCL-1), tumor necrosis factor-alpha (TNF-α) and interferon-gamma (IFN-γ) in samples 6 or 24 h after phosphate-buffered saline (PBS) or lipopolysaccharide (LPS) treatment compared with levels in controls by enzyme-linked immunosorbent assay. Levels of IL-1β, IL-6, TNF-α and CXCL-1 were up-regulated in all conditions compared with the control, while levels of IL-4, IL-5, IL-13 IL-10 and IFN-γ remained similar to those observed in the control. Kruskal–Wallis tests with Dunn’s tests were performed to compare all pairs of columns using GraphPad Prism version 6 (GraphPad Software, CA, USA); *p*-values (bold data highlight significant *p*-values).

**Figure 3 biomedicines-09-00784-f003:**
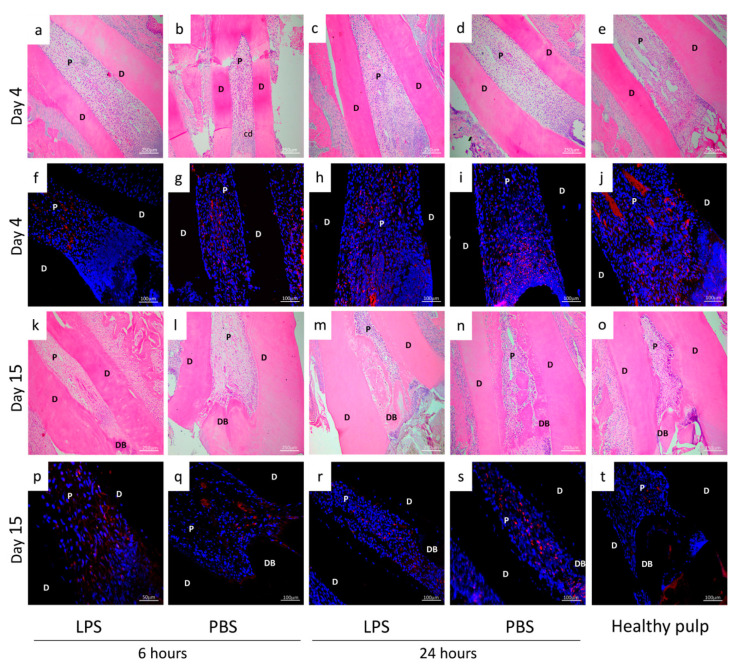
Inflammation analysis by histology and immunochemistry. Hematoxylin-eosin staining (**a**–**e**,**k**–**o**) and immunofluorescence (CD68 label) (**f**,**j**,**p**,**t**) analysis of non-inflamed pulp (healthy pulp) and experimental rat pulpitis induced by phosphate-buffered saline (PBS) or lipopolysaccharide (LPS) treatment observed at 4 and 15 days, after 6 or 24 h of induction. Four days after the pulpotomy, hematoxylin and eosin-staining revealed characteristics of inflammation in all groups. More specifically, CD68 immunohistochemistry highlighted more CD68-positive cells after 24 h (LPS/PBS) (**h**,**i**) than 6 h (LPS/PBS) of induction (**f**,**g**). The labelling is found mainly in the coronal third of the root very close to the cellular degeneration area. Fifteen days after the pulpotomy, the inflammation had resolved in the pulp with 6 h of induction (LPS/PBS) (**p**,**q**) and healthy pulp (**t**). A few macrophage cells remained in the pulp with 24-h induction with PBS/LPS (**r**,**s**). D: dentin, P: pulp, CD: cellular degeneration, DB: dentin bridge.

**Figure 4 biomedicines-09-00784-f004:**
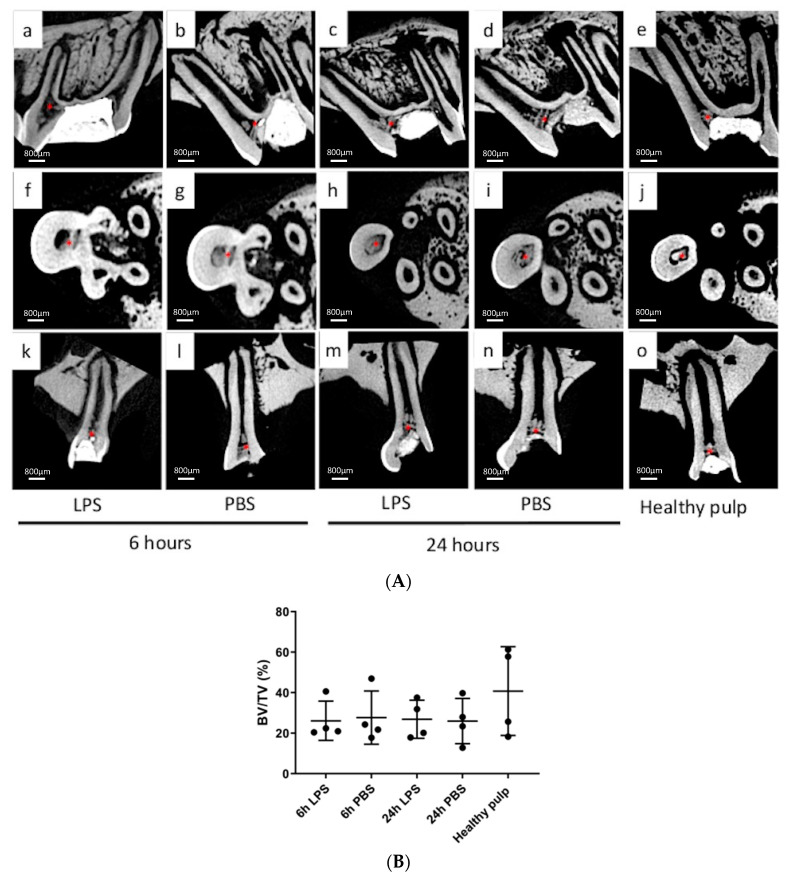
Analysis of mineralized bridge obtained after pulpotomy using tricalcium silicate-based cement by micro-computed tomography. (**A**): The 3D sections focusing on the reparative dentin bridge, 2 weeks after pulpotomy in inflamed teeth (phosphate-buffered saline (PBS) or lipopolysaccharide (LPS)). Thick well-defined calcified tissue (*) was present at the opening of each root, forming a canal obstruction in the control group (**e**,**j**,**o**) and in the 6-h LPS/PBS groups (**a**,**b**,**f**,**g**,**k**,**l**). The mineralized area seen after 24-h induction (**c**,**d**,**h**,**i**,**m**,**n**) looks more heterogenous than after 6-h induction or in healthy pulp. (**B**): Quantification (by Bone volume/Total volume % technique) of the mineralized bridge induced by Biodentine^TM^ 15 days after pulpotomy under different conditions. No significant difference of mineralization was observed between healthy pulp and experimental groups.

**Figure 5 biomedicines-09-00784-f005:**
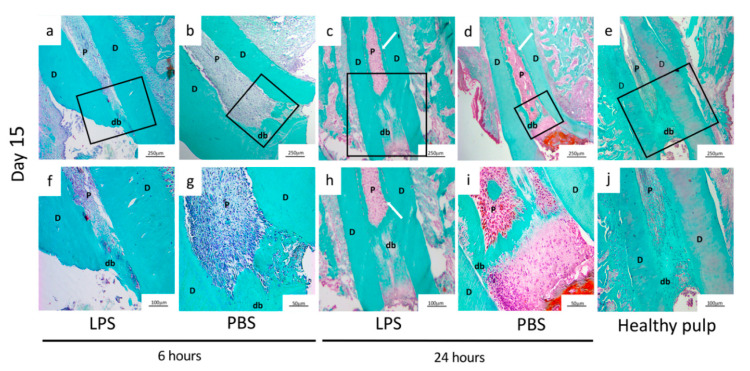
Mineralized bridge evaluation by Masson’s trichrome staining. Masson’s trichrome staining (MTS): (**a**–**e**) low magnification, the box delineates the region shown at higher magnification; (**f**–**j**) analysis of non-inflamed pulp (healthy pulp) and experimental rat pulpitis induced for 6 or 24 h by phosphate-buffered saline (PBS) or lipopolysaccharide (LPS) treatment observed at 15 days. Representative samples of those used to analyze the quality of the dentin bridge under each condition using the scoring table (Table 3). At 15 days after pulpotomy, MTS showed the presence of a mineralized barrier whichever the stimulant and the duration of the inflammation. Differences are observable in the quality of the dentin bridge depending on induction duration. Both the 6-h induction group (**a**,**b**,**f**,**g**) and controls (**e**,**j**) show a mineralized barrier located in the coronal third of the root. The mineralized area of the 24-h condition (**c**,**d**,**h**,**i**) is more extensive along the roots. D: dentin, P: pulp, db: dentin bridge, white arrow: extensive mineralization.

**Figure 6 biomedicines-09-00784-f006:**
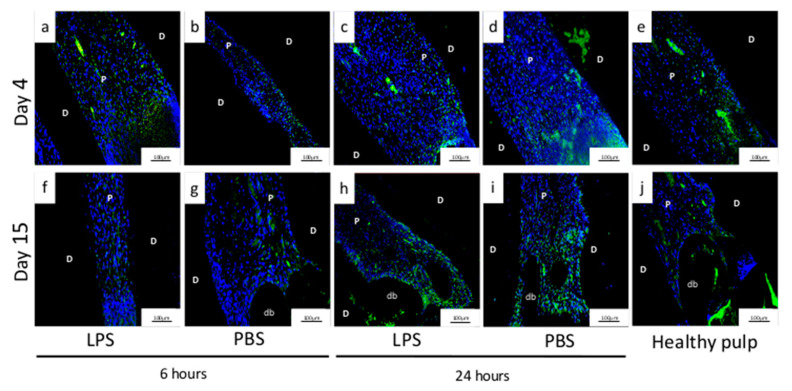
Immunolocalization of DSP-positive cells. Immunolocalization of DSP-positive cells on D4 and D15 after pulpotomy under conditions of pulpitis induced for 6 or 24 h by phosphate-buffered saline (PBS) or lipopolysaccharide (LPS) and in non-inflamed pulp (healthy pulp). More DSP-positive cells are seen on D4 (**a**,**b**) after 6 h of induction with PBS and LPS than on D15 (**f**,**g**), while more are seen on D15 (**h**,**i**) after 24 h of induction with PBS and LPS than on D4 (**c**,**d**). The number of cells appears similar on D4 and D15 in healthy pulp (**e**,**j**). D: dentine, P: pulp, DB: dentine bridge.

## Data Availability

The data presented in this study are available on request from the corresponding author.

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
