# Peer review of "Evaluation of Pulp Repair after Biodentine^TM^ Full Pulpotomy in a Rat Molar Model of Pulpitis"

_biomedicines, 2021, doi:10.3390/biomedicines9070784_

Round 1

Reviewer 1 Report

Please check the MS file. Thank you.

Author Response

This manuscript demonstrates development of a rat model of pulpitis. Also, the manuscript evaluates effects of Biodentine on pulpal healing process, especially focusing on inflammation and reparative dentine formation. The topic is interesting. However, some concerns are remained.

[Authors’ Reply to reviewer’s general comments. ]  We very much appreciate that the reviewer found that “the topic is interesting.”

[ Reviewer comment 1 : In general, please revise the manuscript following the author guideline. For example, Table 1A seems to be not appropriate. Revise the titles of tables as sequential order. ]

Authors’ Reply : We would like to thank the reviewer 1 for this comment in order to improve the manuscript.  The manuscript has been revised following the author guidelines.

[ Reviewer comment 2:  A more  specific  title is advisable. Please consider using “Biodentine” instead of “a bioactive tricalcium silicate cement”. Moreover, revise the title which represents the animal model, since the authors focused on a rat model of pulpitis. ]

Authors’ Reply : A new title was proposed and should be more relevant.

[ Reviewer comment 3:  Please check the authors in the title page. You may be missing some of authors. ]

Authors’ Reply : We would like to thank the reviewer for this comment, authors list has been checked.

[ Reviewer comment 4: Introduction: The flow is quite acceptable. Additionally, cite following reference in the sentence, “Recently, BiodentineTM has been shown to induce the release of large numbers of Ca2+ ions, this  being  reflected  in  altered  intracellular  Ca2+  dynamics,  and,  in  turn,  differential  gene expression, cellular differentiation and mineralization potential of human dental pulp stem cells (hDPSCs) stimulated with TCS-based cements [18]."Nam OH, et al. Differential Gene Expression Changes in Human Primary Dental Pulp Cells Treated with  Biodentine  and  TheraCal  LC  Compared  to  MTA.  Biomedicines.  2020;  8:  445. doi: 10.3390/biomedicines8110445. ]

Authors’ Reply : We would like to thank the reviewer for this very interesting suggestion. This reference was added.

[ Reviewer comment 5: Materials & methods: In  the  section  2.6.,  explain  the  details  of  the  reliability  of  histologic  evaluation.  To  ensure reliability  of  evaluation,  ICC  values  are  required  when  repeated  measurements  of  one-investigator or measurement of multiple investigators are performed. ]

Authors’ Reply : We understand the reviewer’s question but we did not measure the ICC value on the observers, the observers are all scientists dentists used to seeing histological sections of inflammation of the pulp. As the results of the observations were similar, we did not measure the value of CCI.

[ Reviewer comment 6: Results: In the section 3.3., the authors stated that “Less mineralization in inflamed pulp than in healthy pulp. Further, there was a slight difference be-tween the 24-hour and 6-hour induction groups, the mineralization seeming to be less dense in the 24-hour induction group”. It seems to be missing statistical analysis on this matter. Please provide statistical analysis in the Figure 4B.  Or  please  revise  the  manuscript.  The  sentences  cannot  be  justified  without  statistical confirmation. ]

Authors’ Reply : Indeed, the results were not statistically significant, so we revised the manuscript in this sense in the results section.

[ Reviewer comment 7: Results: In the figure 5, some questions are raised. In the figures 5a & 5f, I cannot figure out this phenomenon (see “v” in the following figure). These figures are not similar with the others. Is this some errors during histologic processing or a phenomenon? Please help my inquiry ]

Authors’ Reply : We understand the reviewer's request. Infact, the section plane is not the same as in the other groups; in this group we didn’t succeed to cut the tissue blocks in the same axis as for the other groups.

[ Reviewer comment 8: Discussion: In the section 4.1., the authors demonstrated a new rat model of pulpitis. I found some related articles.Kawashima N, et al. Effect of NOS inhibitor on cytokine and COX2 expression in rat pulpitis. J Dent Res. 2005; 84: 762-7. doi: 10.1177/154405910508400815.Kermeoğlu  F,  et  al. Anti-Inflammatory  Effects  of  Melatonin  and  5-Methoxytryptophol  on Lipopolysaccharide-Induced Acute  Pulpitis  in Rats.  Biomed Res  Int. 2021; 2021: 8884041. doi: 10.1155/2021/8884041.Please, provide the novelty of your model compared to the two articles. ]

Authors’ Reply : We would like to thank the reviewer for this comment. In our study,  we evaluate for the first time for the better of our knowledge that a calcium silicate-based cement induces dentin repair when applied as pulp capping material after pulpotomy in a controlled pulpitis model in rat molars. Our study is novel in that in most cases of experimental induced pulpitis in rats, the studies were realized on incisors as in both suggested studies. However, rodent incisors are continuously erupting teeth with high healing potential after pulp injury and thus removing it from human clinical reality. Regarding rat molar models, most of DP studies are performed after pulpotomy in healthy condition, pulpotomy acting as inflammation producer (Ohkura et al. 2017)(Liu et al. 2015). In comparison with existing studies, our study was intended to be original by evaluating pulp repair after pulpotomy in a rat molar pulpitis model.

Reviewer 2 Report

Dear Authors, the study protocol needs to be  filled with information about why E.coli colonies were used as a source of generation of pulpitis. And why other similiar materials were not used in the study to compare the results. Please clarify in your text this missing information.

Author Response

[Reviewer comment 1: Dear Authors, the study protocol needs to be  filled with information about why E.coli colonies were used as a source of generation of pulpitis. ] 

Authors’ Reply : we added information in discussion section about why E.coli colonies  were used as a source of generation of pulpitis. “ In this study, we evaluated the ability of DP to heal after stimulation with LPS. LPS-induced dental pulp inflammation represents a stable experimental inflammation model for the dynamic observation of the progress of acute and chronic pulpitis[27]. We chose LPS from Escherichia coli for its ability to stimulate the TLR4 receptor, an ability similar to Gram-negative bacteria found in deep caries”.

[Reviewer comment 2: And why other similar materials were not used in the study to compare the results. Please clarify in your text this missing information. ] 

Authors’ Reply . We thank the reviewer for his suggestion. We added this information in the discussion section. “Although there are several cements belonging to the TCS family, we chose to use BiodentineTM for this first part of our research work which aimed to 

Round 2

Reviewer 1 Report

No further concerns are remained. For scientific soundness, please consider ICC values for your further studies. Thanks for your efforts.